# Attitudinal Determinants of Beef Consumption in Venezuela: A Retrospective Survey

**DOI:** 10.3390/foods9020202

**Published:** 2020-02-16

**Authors:** Lilia Arenas de Moreno, Nancy Jerez-Timaure, Jonathan Valerio Hernández, Nelson Huerta-Leidenz, Argenis Rodas-González

**Affiliations:** 1Facultad de Agronomía, Instituto de Investigaciones Agronómicas, Universidad del Zulia, Maracaibo 15205, Venezuela; lilia.arenas@gmail.com; 2Instituto de Ciencia Animal, Facultad de Ciencias Veterinarias, Universidad Austral de Chile, Valdivia 5090000, Chile; 3Universidad Autónoma Chapingo, Km.38.5 Carretera México Texcoco, Chapingo 56230, Mexico; jevh_93@hotmail.com; 4Department of Animal and Food Sciences, Texas Tech University, Box 42141, Lubbock, TX 79409-2141, USA; nelson.huerta@ttu.edu; 5Department of Animal Science, Faculty of Agricultural & Food Sciences, University of Manitoba, Winnipeg, MB R3T 2N2, Canada; Argenis.RodasGonzalez@umanitoba.ca

**Keywords:** eating quality, beef, tenderness, flavor, meat purchase decision-making, meat buying criteria

## Abstract

Consumer surveys were conducted in the Western, Central, and Eastern regions of Venezuela to determine buying expectations, motivations, needs, perceptions, and preferences of beef consumers, and their acceptance of domestic (and foreign) beef, as affected by different intrinsic and extrinsic factors. Data (*n* = 693) were gathered by face-to-face interviews on the way out of fresh markets, butcher stores, supermarkets, and, in some cases, at home by using a 45-question structured questionnaire. Responses were subjected to factorial analysis of correspondence (FA) and hierarchical cluster analysis. From the FA, the first two factors explain 74% of the common variance. Factor 1 comprises intrinsic attributes such as color, smell, tenderness, flavor, juiciness, and freshness; while Factor 2 contains extrinsic attributes, mostly related to the origin. The FA profiling data showed that it is possible to concentrate on the traits that consumers usually use as a criterion to perceive beef quality, and to purchase beef. Using cluster analysis, four groups of consumers were mainly distinguished by region, intrinsic attributes, and credence attributes related to production system, aging, traceability, and hygiene. Results from this study will be helpful in designing strategies for recovering and enhancing the future, domestic beef demand.

## 1. Introduction

Meat is regarded as the most valuable livestock product [1]. Its consumption remains relatively steady in the developed world; however, in developing countries, its annual per capita consumption has doubled since 1980 [1]. This is not the case in Venezuela, where there is currently a strong contraction in the demand for beef due to rampant hyperinflation and a drastic loss of purchasing power [2,3,4]. The Venezuelan beef market has three main marketing channels: “Traditional” (local butcher shops that represents 60 percent of the market), which offers beef and beef products of different qualities, depending on location and the surrounding community’s economic circumstance; “Modern” (supermarkets and medium-sized grocery stores, selling packaged, higher quality that represents 30 percent of the market); and “Industrial” representing 10 percent of the market and is comprised of beef renderers and packers [3,4]. Higher oil prices during the first decade of the millennium allowed for subsidies to import beef products and (or) slaughter cattle from USA, Brazil, Argentina, and Nicaragua; however, in 2003, Venezuela banned all US beef and beef products because of Bovine Spongiform Encefalopathy regulatory concerns [3]. Domestic beef production, imports, and per capita consumption from 2000 to 2019 have been estimated from different private sources [4]. The beef per capita consumption that experienced a sustained rise during the period 2005 to 2011 and reached a historic peak of 24 kg in 2011, fell dramatically to 7 kg (a 70% decrease) in 2018 [3]. Additionally, a lower domestic production (ca. 180,000 MT) and no imports were estimated for 2019, given the deteriorated macroeconomic and market conditions [4]. The population of Venezuela could reach 28.7 million inhabitants by July 2020, a reduction of 7.42% from the population (ca. 31 million) estimated in 2018 [5]. This anticipated population contraction could be explained by the publicly known refugee crisis in the country [6].

Because of the chaotic economic conditions and agri-food situation in Venezuela [2] any consumer survey conducted regarding the current consumers’ habits, needs, quality requirements, and (or) preferences would be of little value for medium- and long-term planning purposes. However, it is clear that, before the ongoing situation of high food insecurity in the country [2], Venezuelans had a high preference for beef as center of the plate [7] and, given the expected axiomatic income elasticity of its demand [8], it can be hypothesized that it may eventually return to its former level if the purchasing power and economic conditions are recovered.

While beef demand is income- and price-sensitive, the purchase decision-making process is also affected by a set of attitudinal and credence factors. Organoleptic attributes perceived by consumers can add or subtract most value to a product, because this set of sensory traits largely determines the acceptance. Perceptions about domestic (and foreign) beef by final consumers are also heavily influenced by their varied socio-cultural background. Perceptions of intrinsic fresh meat attributes (e.g., nutritional value and organoleptic quality) and concerns (ethical quality, food safety, provenance, aging process, feeding, and production technologies, etc.) that influence the purchasing decision-making are poorly documented in Venezuela. Formal surveys designed to discern demand determinants (habits, traditions, preferences, and needs) are not easy to access. This is due to the subject of confidential marketing research for retail corporations or trade associations (e.g., Mercedes Hércules and Asociados, C.A. report for the Venezuelan Beef Council-CONVECAR), cited by Jerez-Timaure et al. [7].

Retrospective data regarding attitudinal responses of beef Venezuelan consumers during the first decade of this century could be helpful in designing strategies for recovering and enhancing the future domestic demand. Hence, the present study aims to (a) learn about attitudinal responses expressed by Venezuelan beef consumers at point of purchase or at home, during the 2007–2008 period, and (b) discover, with their opinions, unmet needs, and opportunities in order to create and capture costumers’ preferences, values, and loyalty to Venezuelan fresh beef products.

## 2. Materials and Methods

### 2.1. Questionnaire Design Mode of Data Collection and Data Classification

The survey’s method was qualitative and consisted in the application of a structured, questionnaire containing 45 questions divided into five sections that included: (1) Socio-demographic characteristics of the participants (Section I, seven questions); (2) Habits in beef consumption (Section II, four questions); (3) Criteria for assessing the quality of raw and cooked meat (intrinsic quality attributes (Section III, 10 questions); (4) Criteria for the evaluation of extrinsic quality attributes (Section IV, 14 questions) and, (5) Motivations for the purchase and/or consumption of beef (Section V, 10 questions).

The questionnaire was administered verbally, in person (face-to-face) using the traditional paper-and-pencil mode to randomly selected households in a total of 693 individuals representing three most populated regions of Venezuela, as follows: (a) Western region with 181 consumers from three cities (Maracaibo, San Francisco, and Cabimas); (b) Central region with 327 consumers from the Caracas, Maracay, and Valencia; and Eastern region with 186 individuals from Barcelona, Puerto La Cruz and Lecherias. Data were collected between 2007 and 2008. It is worth noting that the domestic beef production in 2007 and 2008 was 490,000 MT and 400,000 MT, respectively, whereas the beef imports for the same years were 290,000 MT and 380,000 MT, respectively [3]. It is also noteworthy that the Venezuelan population for 2007 and 2008 was 27.2 million and 27.7 million inhabitants, respectively [9]. The personalized interview method was used at the exit of wet markets, butcheries, supermarkets and, in some cases, at home. This method was considered the most appropriate because of the length of the questionnaire [10]. To ensure the participation of consumers involved in the purchase of food (meats) and groceries, the selection of participants was conditioned on two aspects: (a) to be a beef consumer, (b) to be responsible for the family’s food purchasing-decisions [10,11].

#### 2.1.1. Demographic Characteristics and Beef Consumption Attitudes/Habits

Questions and options to responses related to the socio-demographic characteristics of the participants and habits in beef consumption (Sections I and II) follow: (a) Age (Under 30 years old/30–39 years/40–49 years/50–59 years/60 years. or older/No response); (b) Gender (Male/Female); (c) Civil status (Single/Married/Other/No response); (d) Educational level (Elementary/High School/College/Advanced degree/No response); (e) Occupation (Student/Full-time housewife/Working housewife/Worker/Independent Professional/Full-time employee/Part-time employee/Informal businessman Business owner/Other); (f) Family size (1 Member/2 Members-couple/3 Members/4 Members/More than 4 members); (g) Number of children at home (0/1/2/3/more than 3/No response); (h) Preference for beef (I love it/like it very much/I like it/It does not matter to me/It is not the one I prefer/only if I have no other option/I don’t like it.); (i) Frequency of preparation of meals with beef (Daily/once every 2 or 3 days/Once a week/Once or twice per month/Rarely); (j) Frequency of eating beef as a center of the plate (Daily/once every 2 or 3 days/Once a week/Once or twice per month/Rarely); and (k) Changes in the frequency of consumption due to information received about diet/health issues (Increased consumption/There has been no change/Decreased temporarily but then returned to usual/Decreased consumption/Stop consuming).

#### 2.1.2. Criteria for Assessing the Quality of Raw and Cooked Beef

Questions corresponding to Sections III and IV of the questionnaires focused on aspects related to the criteria for beef quality assessment, which have also been considered in previous research [8,9,10,11,12]. Participants indicated their degree of agreement with the above-mentioned questions/statements using the five-point Likert scale: Definitely yes (5 points); Yes (4 points); Indifferent/Don’t know (3 points); No (2 points); Definitely no (1 point).

##### Intrinsic Attributes

Questions/statements to learn about consumers’ perceptions on intrinsic attributes were: (1) “Beef tenderness is important”, (2) “The color in raw beef is important”, (3) “The smell of raw beef is important”, (4) “The amount of fat present in raw beef is important”, (5) “Preference to buy a leaner beef”, (6) “Freshness (appearance/conservation) is important”, (7) “A highly marbled beef is indicative of good quality” [after ensuring understanding of marbling (i.e., white flecks of intramuscular fat within the lean sections of the beef cut) and noting a chart with color pictures], (8) “The juiciness of cooked beef is important”, (9) “Good flavor is important”, and (10) “Beef with no fat (leaner beef) tastes better” (i.e., more flavorful).

##### Extrinsic Attributes

Questions/statements to learn about consumers’ perceptions on extrinsic attributes follows: (1) “Hygiene is a very important decision factor”, (2) “The type of cut is important”, (3) “The shape and size of the cut are important for cooking”, (4) “The beef aging process is important”, (5) “Are you concerned or would like to know how animals for beef production are fed?”, (6) “Are you concerned that animals intended for meat production will be treated with hormones and/or antibiotics to accelerate their growth?”, (7) “If you are given the option to buy branded beef, even if it was more expensive, would you be willing to pay for it?”, (8) “If you knew the region where the beef was produced, even if it was more expensive, would you be willing to pay for it?”, (9) “If you knew the breed of the animal that produced the meat, even if it were more expensive, would you still be willing to pay it?”, (10) “If beef imported from the USA or Canada -and certified by its country of origin-were regularly offered, would they have a greater consumption than the domestic one?”, (11) “If beef imported from Colombia -and certified by its country of origin-were regularly offered, would it have a greater consumption than the domestic one?”, (12) “If beef imported from other Latin American countries -and certified by its country of origin-were regularly offered, would they have a greater consumption than the domestic one?”, (13) “When purchasing packed meat, do you pay attention for (check out) the packing date?”, and (14) “If you had the opportunity to buy at the same price imported beef or beef produced in Venezuela, would you prefer domestic beef?”.

#### 2.1.3. Consumer’s Buying/Consumption Motivations and Perceptions about Freshness

The set of questions/statements to learn on buying/consumption motivations (section V) follows: (1) “Would you pay a higher price for beef if you’re guaranteed that it will be very tasty (juicy, tender, nicer taste and appearance, altogether?)”, (2) “Would you be willing to pay more for a well-marbled beef?”, (3) “Safety for my family is the most important aspect”, (4) “Would you pay a better price if you are guaranteed that the beef is very safe for your family?”, (5) “Beef with no fat or leaner is good for the human health”, (6) “Would you be willing to pay more for a certified beef free of hormones, antibiotics or other additives?”, (7) “Would you be willing to pay more for beef certified as Natural or Organic?”, (8) “Do you trust the experience of the butcher and his advice at the time of buying beef?”, and (9) “Would you be willing to pay a higher price for beef that allows you to prepare it in an easier and faster way (e.g., seasoned, portioned products, ready to cook). Additionally, participants indicated freshness preferences for buying and/or consuming beef (10) by choosing one of the following seven options: 7 = “Hot beef” knowing that it is derived from today’s slaughtered cattle, 6 = “Very fresh” (beef without refrigeration), 5 = “Refrigerated (cold) meat, with good appearance and color in the showcase”, 4 = “Indifferent/Does not know”, 3 = “Aged beef with good color”, 2 = “Well-aged beef, even if it has lost some color”, and 1 = “Frozen beef”.

### 2.2. Statistical Analyses

All analyses were performed using the software R [12] statistical package. Preliminary and then a Confirmatory Factorial Analysis (FA) of Correspondence [13], and a hierarchical clustering technique were used. The FA helps to determine the combination of attribute levels that are important for consumers, while the cluster technique allows to clustering consumers on several categories, in hierarchical manners. The correlation matrix suggested a high level of collinearity in the group of variables that formed the matrix; Bartlett’s test assessed the homogeneity of variances (α = 0.05), while Univariate (Shapiro-Wilk) and Multivariate (Mardia) normality tests proved that all variables did not have a normal distribution. The FA is part of the multivariate dependency method and does not require assumption tests such as normality; hence, preliminary analyses determined that FA was worthwhile for this data. Frequencies’ distribution and Chi-square analysis for demographics (section I) and beef consumption habits (section II) by region were performed.

The interviewees were classified by means of a cluster analysis of the hierarchical type in order to create exclusive homogeneous classes; with maximum divergence among them, using the k-means method. Frequency analysis was conducted for each cluster or group created. The differences between the groups and the characteristics of the participants were analyzed by Chi-square with the data provided by the frequency analysis.

## 3. Results

### 3.1. Demographics Characteristics and Beef Consumption Habits

Descriptive parameters of demographic characteristics of the participants are shown in Table 1 and differences were established with contingency tables and Chi-square analysis among cities. By analyzing the whole sample structure, most of the consumers were women (67.15%). Regarding occupation, 25.65% expressed that they were in full-time employment, and 20.3% were housewives without formal work. Fifty-two percent of the consumers were in the most productive age (30 to 49 years) and 52.89% of them were graduates from college (Table 1). Fifty-nine percent of the interviewed belonged to a family group of more than 3 people with 2 children or fewer (86%). Significant differences among cities were detected for all variables (*p* < 0.05) but civil status (*p* = 0.14). These findings indicate a difference in the distribution frequencies of almost all demographic traits within the region.

In general terms, the Central region sample is mostly comprised by women; additionally, it is the region that concentrated most of the full-time housewives and the highest educational level percentage (college or advanced degrees). Consumers from the Central region comprise the capital city (Caracas) and two highly populated cites (Maracay and Valencia).

Table 2 shows the consumer´s beef consumption habits of the whole sample and stratified by region. The majority (93.8%) of the interviewed consumers had a high preference for beef, as indicated by their three most frequent responses (“I love it”, “I like it a lot” and “I like it”). It was also observed that most of the interviewees prepared meals with beef as center of the plate every 2 or 3 days (63.0% and 56.9%, respectively); and this trend was very similar across regions; however, the distribution frequency for each question was different among regions (*p* < 0.001).

### 3.2. Factorial Analysis of Correspondence FA

A preliminary FA including all variables/questions from the survey was performed to exclude those variables with little weight on each factor. Four variables/questions with a Kaiser-Meyer-Olkin (KMO) value less than 0.6 were excluded from further analysis, these were: marbling (as an intrinsic trait), butcher trustworthiness, freshness preferences for buying and/or consuming beef and preference for ready-to-eat meat. The confirmatory FA analysis showed that the most significant variables/questions were simplified into three (3) factors according to the R-square value; the first two FAs gathered the traits explaining most of the total variability, and the first two factors explain 74% of the common variance (Table 3).

Projections of main variables/questions answered by Venezuelan consumers on intrinsic and intrinsic attributes and buying/consumption motivations (in the FA1 and FA2 axes) are shown in Figure 1. The first factor (FA1) is mainly defined by the importance of “intrinsic attributes” such as tenderness, smell, flavor, color, juiciness, and freshness. The second factor (FA2) shows that extrinsic attributes like origin (traceability, breed information) and beef brand are important. The third factor (FA3) (not shown in Figure 1) highlights the consumer perception that imported beef is better than its domestic counterpart as an extrinsic attribute of beef quality.

In Figure 2, intrinsic (FA1) and extrinsic attributes (FA2) are represented by the two-dimensional space defined by these factors or axes, by region. Preferences and motivations for purchasing beef based on perceived freshness were different (*p* < 0.05), and more noticeable between consumers from Western and Eastern regions. However, consumers from the Central region showed wider preferences and motivations that cover those expressed by counterparts in both Western and Eastern regions. Eastern consumers were more concentrated and gave more importance to origin and production traits (breed type, animal feeding, use of hormones). Additionally, they recognized the importance of aging to improve beef quality; therefore, they were willing to pay more for branded beef, traceability, and tastier meat. Consumers from the Western region gave more importance to intrinsic beef attributes (tenderness, flavor, juiciness, smell, color, and freshness), while consumers in the Central region showed a broader spectrum of preferences and motivations to buy beef.

### 3.3. Cluster Analysis Characteristics of Different Types of Consumers

Cluster analysis allowed identifying the conformation of four groups of consumers that showed to have a small as possible of internal variants, the advantages of this method are using variance analysis to check the distance between clusters. Figure 3 shows the hierarchical clustering of the four groups of consumers and the Euclidean distance among them. For instance, G1 (showed by red color) and G4 (purple color) were the closest groups, whereas G2 (green color) and G3 (blue color) resulted to be the most divergent ones.

The frequencies for each cluster (group of consumers) variables/question that resulted with *p* < 0.05 in the Chi-square test and resulted with the highest eigenvalue in the FA analysis, are depicted in Table 4 (intrinsic attributes), Table 5 (extrinsic attributes), and Table 6 (consumer´s buying/consumption motivations and perceptions.

It is worthwhile to highlight that most of the intrinsic attributes were very important for all consumers interviewed; however, consumers from G3 group were more sensitive to these attributes than those from G1 group. All consumers grouped in G1 were from the Eastern region (*n* = 138) and represent 20% of the total data. Consumers from G1 assigned greater importance to extrinsic factors such as hygiene and beef aging; however, they also would like to know about the origin, feeding practices and breed of the animals; hence, they are willing to pay more for higher quality, freedom of hormones, and certified organic beef. The G2 group (*n* = 390) represents 56.3% of the total consumers, mostly from the central region (76.77%), who were more sensitive to intrinsic attributes, mainly freshness, tenderness, flavor, and color and less worry about extrinsic attributes like origin, breed and feeding, among others. The G3 group considers as very important, the beef intrinsic quality traits (tenderness, color, smell, freshness, and juiciness); and represents 20% of the total data (n = 138), most of them (85%) from the Western region. Lastly, G4 was the smallest group comprised of only 24 consumers, all from the Western region (*n* = 27). Consumers in G4 did not give importance to extrinsic factors such as beef aging, origin, breed, or feeding practice but were willing to pay more for safe, good-quality beef; they also preferred domestic beef rather than the imported ones.

## 4. Discussions

Consumers are the last segment of the beef chain, and having their expectations met is an important part of their satisfaction and shopping behavior. It is, therefore, important to understand the factors affecting consumer behavior [14]. On the other hand, consumer perception is generally dynamic, and its development needs to be monitored continuously as one of the information resources for decision-making in all the food chain. Villalobos et al. [15] stated that Chilean consumers can modify their choices and habits, prioritizing quality attribute differentiators over economic aspects at the time of purchase.

Few studies have evaluated the demand determinants based on beef quality attributes in developing countries. Castillo and Carpio [16] conducted a survey of 574 households in Ecuador and reported that they have a positive interest in all credence attributes (i.e., sanitary control, aging, animal welfare and traceability). In Brazil, Giacomazzi et al. [17] reported that the most valued attributes for beef consumers are appearance, price, and type of cut; while brand and certifications had little relevance as purchase-decision criteria.

To get a broader representation of Venezuelan consumers, surveys were performed in the regions that concentrate most of the population, including Caracas, the capital city of the country. From the total sample of consumers (*n* = 693), most of them (67.1%) were women. This finding concurs with those of Segovia et al. [18], Pierce-Colfer et al. [19], Mahbubi et al. [20] and Castillo and Carpio [16], supporting the argument that women dominate the decision-making for beef consumption in their family.

Educational status affects food consumption patterns, which reflects more nutrition and dietetic knowledge, and health-food perceptions, as stated by Bhurosy and Jeewon [21]. For Krystallis and Arvanitoyanis [10], age and education are the most important socio-demographic characteristics that influence consumers' attitudes towards meat. Older consumers associate meat purchases to the risk concept, mainly due to perceived diet-health issues.

Most of the consumers interviewed herein have families comprised of four or more members (Table 1). Yee et al. [22] stated that smaller families tend to be well-nourished and healthier because they have a broader opportunity to access sources of animal protein. Bonny et al. [23] highlighted other demographic factors such as income, gender, employment status, occupation, city or region, number of children and adults at home; while the frequency of beef consumption and degree of doneness had little effect on beef preferences.

Participants in this survey showed a relatively high preference for beef. The majority (93.8%) responded that they like beef or like it very much, and more than 50% of the responders consumed beef at least every 2 or 3 days. High frequencies of beef consumption have also been reported by the majority (>50%) of responders in other Latin American countries like Mexico (i.e., 75%, by Vilaboa-Arroniz et al. [24], and 95% by Ngapo et al. [25]), Ecuador (79%, by Castillo and Carpio [16]), Costa Rica (91%, by CORFOGA, [26]), and Chile (82%, [27]). Additionally, in Australia, Ardeshiri and Rose [28] reported from a survey of 1002 Australian residents that the majority of respondents (51.8%) purchase beef once a week.

Intrinsic and extrinsic beef attributes are important determinants of consumer´s demand for beef; however, the relative importance of these factors varies from country to country [16] or from city to city (in the same country), as it was detected in the present study. Intrinsic cues relate to physical aspects of the product (e.g., color, shape, appearance, etc.) whereas extrinsic ones relate to the product but are not physically part of it (brand, origin, store, packaging, type of cut, production information, etc.).

From the preliminary FA, four variables/questions with a low KMO value were excluded from further FA analysis, namely marbling (as an intrinsic trait and willingness), butcher trustworthiness, freshness preferences for buying and/or consuming beef, and preference for ready-to-eat meat. These results showed that these attributes were not as important for Venezuelan consumers.

Among the intrinsic attributes, marbling, and color stand out as the most important beef quality traits in other surveys [29]. Marbling is well known by consumers in countries with long-standing appreciation for beef quality, and it is frequently used as an intrinsic clue (and consumers are willing to pay for highly marbled beef); particularly in those countries with well-established beef carcass quality systems or meat standards (i.e., USA, Canada, Australia, Japan) [30]. Additionally, it is widely accepted that the use of early-maturing *Bos taurus* genetics, castration and grain feeding favors deposition of intramuscular fat [31]. However, in Venezuela, most of the cattle are genetically composed by *Bos indicus* types (mainly Brahman straight-breeds and crossbreds) because of their adaptability to harsh, tropical conditions. Brahman-influenced breed-types have been characterized in many studies as having lower intramuscular fat content (and hence, lower marbling scores) as compared to *Bos taurus* breed types [31]. Additionally, the beef cattle in Tropical America are mostly raised and fattened on pastures with little supplementation (i.e., lower energy diets), which contributes to a poor marbling performance. Moreover, the predominant slaughter cattle in Venezuela are intact males whose carcasses are leaner (with lower marbling scores) than castrates and cull females. For instance, Jerez and Huerta [32] described beef carcasses derived from grass-fed bulls with low marbling scores (fluctuating between “Traces” and “Slight” amounts). This observation can explain why most Venezuelan consumers are not familiar with marbling or do not recognize it as an important quality trait/clue.

Preference for frozen vs. refrigerated beef or ready-to-eat beef was irrelevant. Indifference (or lack of understanding) to ready-to-eat meals may be explained for the entrenched custom of many Venezuelans to prefer to eat at home. On the other hand, extrinsic attributes such as butcher trustworthiness were disregarded by consumers. This may be due to the growth of self-service store (without personalized attention) in the cities under study. It should be noted that many of the surveys were carried out at the exit of this type of store. Additionally, there are many anecdotes of bad experiences dealing with butchers, which leads to a bad opinion of these merchants. Previous experiences could modify the quality expectations of meat products [29].

The confirmatory FA revealed that intrinsic attributes such as tenderness, color, juiciness, smell, flavor, and freshness were deemed important by most consumers; however, cluster analysis revealed that intrinsic attributes, such as tenderness, color, flavor, and smell were more important for G3 than for the G1 group of consumers. The brilliant red color is desirable for most consumers and it determines the purchasing decision, probably because consumers perceive discoloration as an indicator of spoilage [33]. In other countries, consumers also relate red–purple color with freshness and brown color with a lack of freshness [11,14,27].

Most of the consumers from this survey (G3, G2, and G4) highly regarded tenderness as an important attribute for beef preference, and interestingly, they acknowledged more the beneficial effect of aging on beef quality/tenderness than marbling. The latter perception is supported by science because Khan et al. [34] and many others demonstrated that beef aging positively affects the final texture, juiciness, and aroma developed through the major biochemical processes-proteolysis, lipolysis, and oxidation.

Several extrinsic factors related to origin (i.e., breed information, animal feeding, traceability, etc.) were important to Venezuelan consumers. According to the FA analysis, these attributes are regarded as some of the most important considerations for explaining differences among consumers. The G1 group was more sensitive to extrinsic attributes of beef; generally, consumers preoccupied about origin of beef also have a higher preference for healthier beef (free of hormones, additives or certified as natural or organic). Bernués et al. [35] pointed out that animal feeding and the provenance are the most important extrinsic attributes for European consumers and supported that origin of meat has also been associated with meat safety. In France, a survey with 625 consumers [36] revealed that consumers are willing to pay for meat products with guaranteed attributes such as labels, traceability, tenderness, and certifications. Meanwhile, in Latin American (Chile), Villalobos et al. [15] found that quality assurance was the most important attribute in the consumer's beef purchase decision process, followed by country of origin, production system, and price.

Perceptions of beef quality and motivations to buy beef have shown to be different among regions. However, from a marketing standpoint, results from the Central region—which included the most populated cities—indicate that its attitudinal pattern could be representative of the rest of the country. However, cluster analysis provides a more precise consumer segmentation and allowed to determine the demographics and preferences profile of each group. For instance, cluster analysis segregated consumers from the Eastern region in one group (G1); and this group had a particular interest in origin and traceability, perhaps because most (73%) of them had some college education and 55.07% were independent professionals or full-time employees; hence, they seem to be better informed about beef production. On the other hand, the majority of the G2 consumers were from the Central region, with a high-school educational level (41.5%), and most of them were full-time housewives. Most of the consumers from the Western region were concentrated in the G4 group, who dismissed extrinsic attributes over intrinsic ones.

## 5. Conclusions

Two factors could explain the higher proportion of heterogeneity in this sample of Venezuelan beef consumers. Intrinsic attributes such as tenderness, color, smell, flavor, freshness, and juiciness; as well as extrinsic attributes such as aging, hygiene, origin, breed, and animal feeding information were important for Venezuelan consumers.

The FA analysis of the profiling data showed a distinct location for each region in the multivariate space and four groups of consumers were defined by cluster analysis, which demonstrated that the relative importance that consumers are giving to the different beef attributes, as determinants of its purchase and consumption, varies widely between regions, and, in turn, depends on the educational level, occupation and other sociodemographic characteristics.

This reality constitutes a great opportunity for consumer-driven product development and further market segmentation and indicates that consumers need to be well informed and educated on quality and food safety matters, by means of trustworthy sources of information.

## Figures and Tables

**Figure 1 foods-09-00202-f001:**
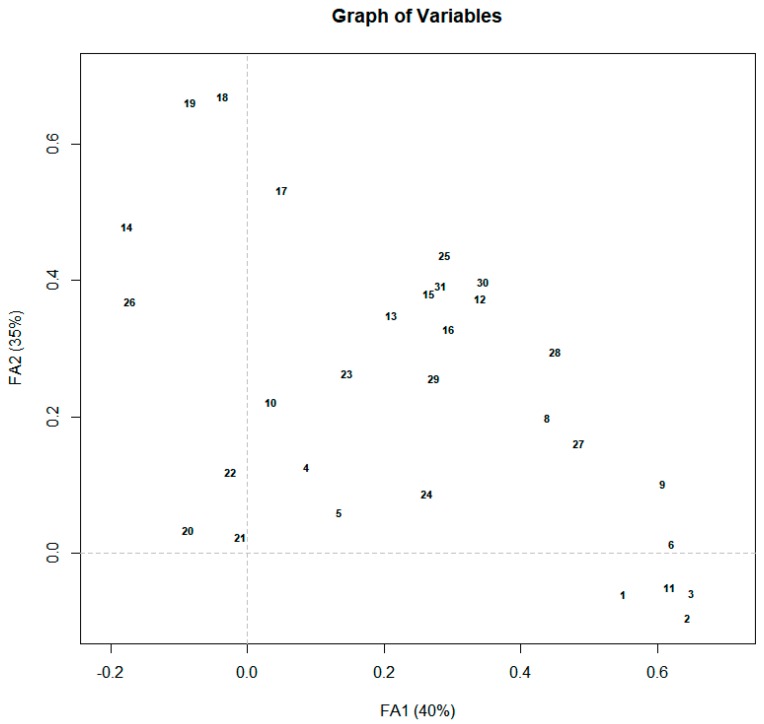
Projection of main variables/questions about consumers’ perceptions of intrinsic and intrinsic attributes and consumers buying/consumption motivations in the FA1 and FA2 axes. Variables/question are shown as numbers as follow: Intrinsic attributes: Beef tenderness is important (1); The color in raw beef is important (2); The smell of raw beef is important (3); The amount of fat in raw beef is important (4); Leaner beef taste better (5); Freshness is important (6); A highly marbled beef is indicative of good quality (7); The juiciness is important (8); Good flavor is important (9); Leaner beef taste better (10). Extrinsic attributes: Hygiene is very important (11); The type of cut is important (12); The shape and size of the cut are important (13); Beef aging is important (14); Animal feeding is important (15); Use of hormones/antibiotics is important (16); Beef brand is important (17); Traceability is important (18); Breed information is important (19); Beef imported from the USA or Canada would have a greater consumption than the domestic one? (20); Beef imported from Colombia would have a greater consumption than the domestic one? (21); Beef imported from other Latin American countries would have a greater consumption than the domestic one? (22); Label information is important (23); Domestic beef is better than the imported one (24). Consumer’s purchasing motivations: Would you pay a higher price for tasty beef (25); Would you be willing to pay more for a well-marbled beef? (26); Safety for my family is the most important aspect (27); Would you pay more for safe beef? (28); Leaner beef is good for human health (29); Would you be willing to pay more for a certified beef free of hormones, antibiotics or other additives? (30); Would you be willing to pay more for beef certified as Natural or Organic? (31); Do you trust the experience of the butcher (32); Freshness preferences for buying and/or consuming beef (33); Would you be willing to pay a higher price for beef that allows you to prepare it in an easier and faster way (34).

**Figure 2 foods-09-00202-f002:**
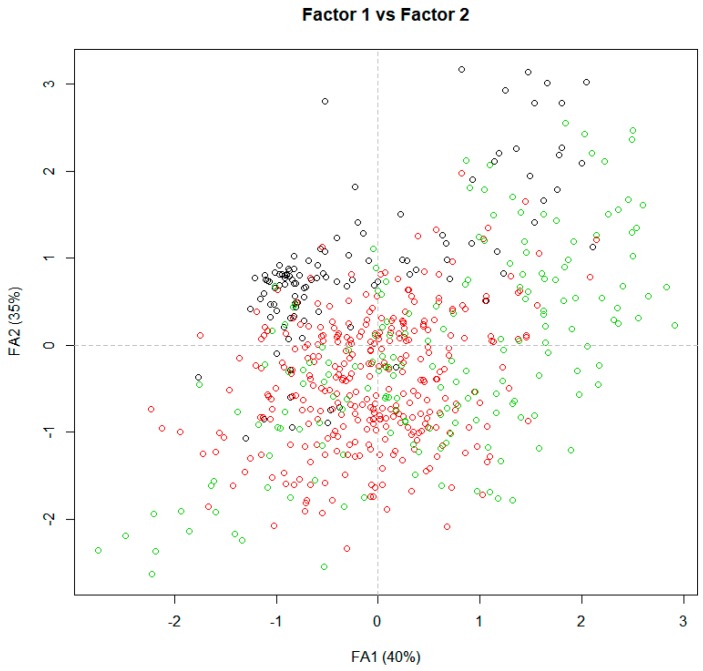
Projection of the Central (green), Western (black), and Eastern (red) regions for the first and second factors (FA1 and FA2, respectively).

**Figure 3 foods-09-00202-f003:**
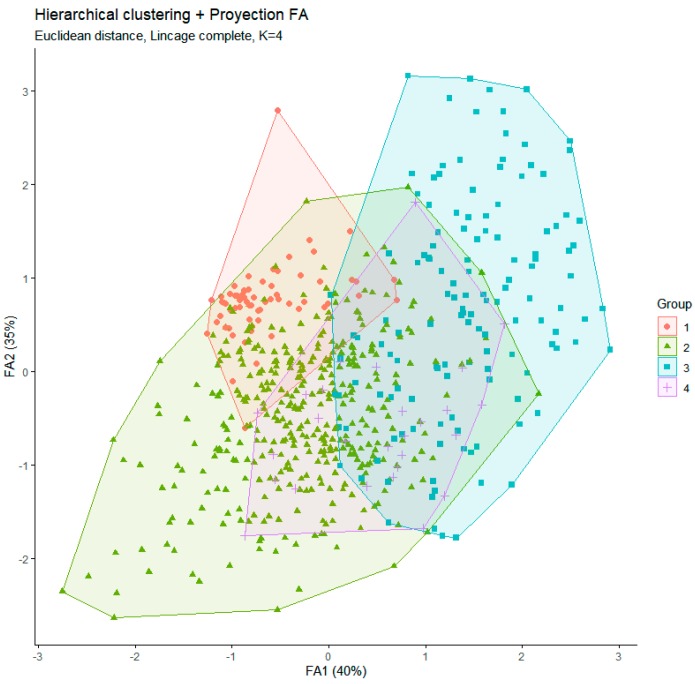
Hierarchical clustering and group of consumers projected in the FA1 and FA2 axes.

**Table 1 foods-09-00202-t001:** Demographic characteristics for the whole sample and by region.

Questions	Optional Responses	Total, %	Central Region (*n* = 327)	Western Region (*n* = 181)	Eastern Region (*n* = 186)	*p*-Value
Age	≤29 years old	15.27	6.8	5.3	3.2	0.004
	30–39 years old	25.79	10.4	8.4	7.1	
	40–49 years Old	26.66	12.4	5.8	8.5	
	50–59 years Old	20.17	9.8	4.6	5.8	
	≥60 years	9.37	5.6	1.7	2.0	
	No response	2.74	2.2	0.3	0.3	
Gender	Male	32.85	13.3	8.6	11.0	0.01
	Female	67.15	33.9	17.4	15.9	
Civil Status	Single	29.97	15.4	8.2	6.3	0.14
	Married	58.50	26.9	14.0	17.6	
	Other	11.10	4.5	3.7	2.9	
	No response	0.43	0.3	0.1	0.0	
Educational level	Elementary	9.51	5.5	3.7	0.3	<0.001
	High School	36.02	18.9	11.8	5.3	
	College	46.69	18.6	8.8	19.3	
	Advanced degree	6.20	2.9	1.4	1.9	
	No response	1.59	1.3	0.3	0.0	
Occupation	Student	6.20	3.0	2.2	1.0	<0.001
	Full-time housewife	20.32	12.7	5.9	1.7	
	Working housewife	9.22	1.9	2.4	4.9	
	Worker	4.47	2.4	1.3	0.7	
	Independent Professional	16.28	6.5	2.4	7.3	
	Full-time employee	25.65	10.7	6.9	8.1	
	Part-time employee	3.75	0.3	1.4	2.0	
	Informal businessman	8.07	5.5	2.3	0.3	
	Business owner	1.59	0.6	0.4	0.6	
	Others	4.47	3.6	0.7	0.1	
Family Size	1 member	4.47	2.0	0.6	1.9	<0.001
	2 members-couple	11.82	4.9	2.6	4.3	
	3 members	24.78	8.1	6.2	10.5	
	4 members	24.78	12.0	5.8	7.1	
	More than 4 members	34.15	20.2	11.0	3.0	
Number of children at home	0	36.17	19.6	8.2	8.4	<0.001
	1	28.10	9.7	8.4	10.1	
	2	21.90	9.7	5.8	6.5	
	3	8.21	5.3	1.6	1.3	
	More than 3	4.90	2.7	1.7	0.4	
	No response	0.72	0.1	0.4	0.1	

**Table 2 foods-09-00202-t002:** Beef consumption habits for the whole sample and by region.

Consumption Habits	Optional Responses	Total, %	Central Region (*n* = 327)	Western Region (*n* = 181)	Eastern Region (*n* = 186)	*p*-Value
Preference for beef	I love it	15.13	7.9	4.9	2.3	<0.001
	I like it very much	25.22	5.0	5.9	14.3	
	I like it	37.90	21.5	8.5	8.4	
	It does not matter to me/It is not the one I prefer	9.37	5.2	3.2	1.0	
	It is not the one I prefer	9.22	6.2	2.6	0.4	
	Only if I have no other option	2.74	1.6	0.9	0.3	
	I do not like it	0.43	0.1	0.1	0.1	
Frequency of preparations od meals with beef	Daily	9.37	5.3	3.3	0.7	<0.001
	Every 2 or 3 days	62.97	26.8	14.7	21.5	
	Once a week	19.74	11.1	5.5	3.6	
	1 or 2 times a month	3.60	2.4	0.6	0.6	
	Rarely	4.32	1.4	2.4	0.4	
Frequency of eating beef as a center of the plate	Daily	7.78	4.8	2.7	0.3	<0.001
	Every 2 or 3 days	56.92	27.1	13.5	16.3	
	Once a week	26.22	11.1	6.1	9.1	
	1 or 2 times a month	5.76	3.2	2.0	0.6	
	Rarely	3.31	1.0	1.7	0.6	
Changes in the frequency of consumption due to information received about diet/health issues	Increased consumption	2.02	1.6	0.3	0.1	<0.001
	There has been no change	61.53	32.3	13.0	16.3	
	Decreased but then returned to usual	13.26	1.3	4.3	7.6	
	Decreased consumption	22.19	11.8	7.6	2.7	
	Stop consuming	1.01	0.1	0.9	0.0	

**Table 3 foods-09-00202-t003:** Results of the confirmatory factorial analysis for questions addressing intrinsic attributes, extrinsic attributes, and consumer purchasing motivations.

Question/Variable	FA1	FA2	FA3
Intrinsic Attributes			
Beef tenderness is important	**0.52**	−0.06	−0.04
The smell of raw beef is important	**0.61**	−0.06	−0.07
Flavor is important	**0.58**	0.10	−0.13
The color in raw beef is important	**0.61**	−0.09	−0.06
Leaner beef taste better	0.00	0.22	−0.02
The amount of fat is important	0.05	0.12	−0.09
Juiciness is important	**0.41**	0.20	−0.14
Preferences for leaner beef	0.10	0.06	0.02
Freshness (appearance/conservation) is important	**0.59**	0.01	−0.17
Extrinsic Attributes			
Hygiene is a very important	**0.59**	−0.05	−0.10
Type of cut is important	0.30	0.37	−0.16
Shape and size of cut is important	0.17	0.35	−0.15
Beef aging is important	−0.21	**0.48**	0.01
Animal feeding is important	0.23	0.38	−0.03
Use of hormones/antibiotics is important	0.30	0.36	0.05
Beef brand is important	0.01	**0.53**	0.10
Traceability is important	−0.07	**0.67**	0.09
Breed information is important	−0.12	**0.66**	0.05
Beef imported from the USA or Canada would have a greater consumption than the domestic one?	−0.12	0.33	**0.66**
Beef imported from Colombia would have a greater consumption than the domestic one?	−0.05	0.02	**0.86**
Beef imported from other Latin American countries would have a greater consumption than the domestic one?	−0.06	0.12	**0.78**
Label information is important	0.11	0.26	−0.12
Domestic beef is better than the imported one	0.23	0.09	−0.34
Consumer’s Purchasing Motivations			
Would you pay more for a tasty beef?	0.25	**0.44**	0.04
Safety for my family is the most important aspect	**0.45**	0.16	−0.12
Would you pay a better price if you are guaranteed that the beef is very safe for your family?	0.41	0.29	−0.04
Leaner beef is healthier	0.24	0.26	−0.12
Would you be willing to pay more for a beef certified as free of hormones, antibiotics or other additives?	0.31	**0.40**	0.05
Would you be willing to pay more for beef certified as Natural or Organic?	0.28	0.36	0.17
Would you be willing to pay more for a well-marbled beef?	−0.21	0.37	0.06
Variance by factor (%)	40	35	26
Cumulative variance (%)	40	74	100
R-square	0.82	0.81	0.84

The root means square of the residuals (RMSR) is 0.07. Bartlett´s K-squared = 2844.4, df = 33, *p* < 0.001. Numbers highlighted in bold are the highest standarized factors loading (≥0.40).

**Table 4 foods-09-00202-t004:** Intrinsic attributes with significant differences (*p* < 0.05 *) in the groups of consumers identified by cluster analysis.

Question/Variable	G 1 (*n* = 138)		G 2 (*n* = 390)		G 3 (*n* = 138)		G 4 (*n* = 27)	
	*n*	%	*n*	%	*n*	%	*n*	%
Beef Tenderness Is Important								
1	0	0.00	1	0.26	0	0.00	0	0.0
2	0	0.00	6	1.54	2	1.45	0	0.0
3	0	0.00	8	2.05	3	2.17	0	0.0
4	137	99.28	204	52.31	24	17.39	20	74.1
5	1	0.72	171	43.85	109	78.99	7	25.9
Color of Raw Beef Is Important								
1	0	0.00	0	0.00	0	0.00	0	0.0
2	0	0.00	6	1.54	0	0.00	0	0.0
3	0	0.00	10	2.56	1	0.72	0	0.0
4	133	96.38	207	53.08	23	16.67	14	51.9
5	5	3.62	167	42.82	114	82.61	13	48.1
Smell of Raw Beef Is Important								
1	0	0.00	1	0.26	0	0.00	0	0.0
2	0	0.00	2	0.51	0	0.00	0	0.0
3	0	0.00	5	1.28	1	0.72	0	0.0
4	117	84.78	197	50.51	20	14.49	9	33.3
5	21	15.22	185	47.44	117	84.78	18	66.7
Freshness (Appearance/Conservation) Is Important								
1	1	0.72	0	0.00	0	0.00	0	0.0
2	0	0.00	2	0.51	0	0.00	0	0.0
3	0	0.00	0	0.00	1	0.72	0	0.0
4	135	97.83	307	78.72	22	15.94	9	33.3
5	2	1.45	81	20.77	115	83.33	18	66.7
Juiciness of Cooked Beef Is Important								
1	0	0.00	1	0.26	0	0.00	0	0.0
2	0	0.00	15	3.85	5	3.62	0	0.0
3	0	0.00	19	4.87	2	1.45	0	0.0
4	134	97.10	308	78.97	47	34.06	14	51.9
5	4	2.9	47	12.05	84	60.87	13	48.10
Good flavor is Important								
1	0	0.00	1	0.26	0	0.00	0	0.0
2	0	0.00	2	0.51	0	0.00	0	0.0
3	0	0.00	5	1.28	0	0.00	0	0.0
4	131	94.93	285	73.08	27	19.57	2	7.4
5	7	5.07	97	24.87	111	80.43	25	92.6

Participants indicated their degree of agreement using the 5-point Likert scale: 1: Definitely no; 2: No; 3: Indifferent/Don’t know; 4: Yes; 5: Definitely yes. * obtained from Chi-squared test.

**Table 5 foods-09-00202-t005:** Extrinsic attributes with significant differences (*p* < 0.05 *) in the groups of consumers identified by cluster analysis.

Question/Variable	G 1 (*n* = 138)		G 2 (*n* = 390)		G 3 (*n* = 138)		G 4 (*n* = 27)	
	*n*	%	*n*	%	*n*	%	*n*	%
Hygiene Is a Very Important Factor								
1	0	0.00	0	0.00	0	0.00	0	0.0
2	0	0.00	1	0.26	0	0.00	0	0.0
3	0	0.00	0	0.00	0	0.00	0	0.0
4	123	89.13	179	45.90	7	5.07	2	7.4
5	15	10.87	210	53.85	131	94.93	25	92.6
The Beef Aging Process Is Important								
1	0	0.00	11	2.82	12	8.70	6	22.2
2	1	0.72	86	22.05	24	17.39	11	40.7
3	4	2.90	136	34.87	27	19.57	9	33.3
4	131	94.93	148	37.95	41	29.71	1	3.7
5	2	1.45	9	2.31	34	24.64	0	0.0
Are you Concerned or Would Like to Know How Animals for Beef Production are Fed?								
1	0	0.00	2	0.51	1	0.72	5	18.5
2	0	0.00	43	11.03	6	4.35	11	40.7
3	1	0.72	24	6.15	5	3.62	3	11.1
4	132	95.65	285	73.08	57	41.30	5	18.5
5	5	3.62	36	9.23	69	50.00	3	11.1
Are You Concerned that Animals Intended for Meat Production Are Treated with Hormones and/or Antibiotics to Accelerate Their Growth?								
1	0	0.00	4	1.03	2	1.45	0	0.0
2	0	0.00	35	8.97	5	3.62	0	0.0
3	1	0.72	23	5.90	3	2.17	1	3.7
4	128	92.75	280	71.79	53	38.41	18	66.7
5	9	6.52	48	12.31	75	54.35	8	29.6
If You Are Given the Option to Buy Branded Beef, Even If it Was More Expensive, Would You be Willing to Pay for It?								
1	0	0.00	2	0.51	7	5.07	2	7.4
2	1	0.72	101	25.90	26	18.84	7	25.9
3	1	0.72	26	6.67	10	7.25	8	29.6
4	135	97.83	246	63.08	41	29.71	7	25.9
5	1	0.72	15	3.85	54	39.13	3	11.1
If You Knew the Region Where the Beef Was Produced, Even If It Was More Expensive, Would You be Willing to Pay for It?								
1	0	0.00	3	0.77	8	5.80	2	7.4
2	1	0.72	168	43.08	35	25.36	13	48.1
3	3	2.17	50	12.82	7	5.07	10	37.0
4	133	96.38	163	41.79	55	39.86	1	3.7
5	1	0.72	6	1.54	33	23.91	1	3.7
If You Knew the Breed of the Animal that Produced the Meat, Even If It Were More Expensive, Would You Still be Willing to Pay It?								
1	0	0.00	2	0.51	12	8.70	3	11.1
2	1	0.72	161	41.28	32	23.19	12	44.4
3	4	2.90	57	14.62	11	7.97	10	37.0
4	131	94.93	162	41.54	56	40.58	1	3.7
5	2	1.45	8	2.05	27	19.57	1	3.7
If you had the opportunity to buy at the same price imported beef vs. beef produced in Venezuela, would you willing to pay for it?								
1	0	0.00	1	0.26	0	0.00	0	0.0
2	16	11.59	38	9.74	7	5.07	5	18.5
3	7	5.07	36	9.23	5	3.62	10	37.0
4	113	81.88	284	72.82	38	27.54	12	44.4
5	2	1.45	31	7.95	88	63.77	0	0.0

Participants indicated their degree of agreement using the five-point Likert scale: 1: Definitely no; 2: No; 3: Indifferent/Don’t know; 4: Yes; 5: Definitely yes. * obtained from Chi-squared test.

**Table 6 foods-09-00202-t006:** Consumer’s buying/consumption motivations and perceptions with significant differences (*p* < 0.05 *) in the groups of consumers identified by cluster analysis.

Variable	G 1 (*n* = 138)		G 2 (*n* = 390)		G 3 (*n* = 138)		G 4 (*n* = 27)	
	*n*	%	*n*	%	*n*	%	*n*	%
Would you Pay a Higher Price for Beef If You Are Guaranteed that It Will be Very Tasty (Juicier, Tender, Nicer Taste and Appearance, Altogether?								
1	0	0.00	3	0.77	2	1.45	0	0.0
2	0	0.00	46	11.79	5	3.62	0	0.0
3	1	0.72	23	5.90	3	2.17	0	0.0
4	135	97.83	295	75.64	53	38.41	22	81.5
5	2	1.45	23	5.90	75	54.35	5	18.5
Safety for My Family Is the Most Important Aspect								
1	0	0.00	0	0.00	0	0.00	0	0.0
2	0	0.00	17	4.36	0	0.00	0	0.0
3	0	0.00	10	2.56	0	0.00	0	0.0
4	135	97.83	272	69.74	22	15.94	10	37.0
5	3	2.17	91	23.33	116	84.06	17	63.0
Would You be Willing to Pay More for a Certified Beef Free of Hormones, Antibiotics or Other Additives?								
1	0	0.00	0	0.00	0	0.00	0	0.0
2	0	0.00	36	9.23	12	8.70	0	0.0
3	0	0.00	21	5.38	6	4.35	0	0.0
4	133	96.38	302	77.44	42	30.43	20	74.1
5	5	3.62	31	7.95	78	56.52	7	25.9
Would You be Willing to Pay More for Beef Certified as Natural or Organic?								
1	0	0.00	2	0.51	1	0.72	0	0.0
2	10	7.25	63	16.15	19	13.77	0	0.0
3	1	0.72	35	8.97	10	7.25	0	0.0
4	123	89.13	262	67.18	39	28.26	17	63.0
5	4	2.90	28	7.18	69	50.00	10	37.0

Participants indicated their degree of agreement using the five-point Likert scale: 1: Definitely no; 2: No; 3: Indifferent/Don’t know; 4: Yes; 5: Definitely yes. *obtained from Chi-squared test.

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
