# Peer review of "Attitudinal Determinants of Beef Consumption in Venezuela: A Retrospective Survey"

_foods, 2020, doi:10.3390/foods9020202_

Round 1
Reviewer 1 Report
This manuscript is very clear, well written and interesting. I have only minor comments.
Lines 37-44. Please indicate the size of the market by providing, for instance, the number of people living in Venezuela, and the amount of beef produced within the country and the amount of imported beef.
The English language should be checked. For instance, line 87, it should be “selection of participants”. Line 251, it should be intrinsic attributes (with s).
Please could you indicate the difference between Conjoint Analysis and Factorial Analysis of Correspondence (which I would have recommended)?
Line 205: please define KMO.
I think it is very surprising that marbling and butcher trustworthiness were removed as they are known as important traits for consumers. See for instance Polkinghorne RJ, Thompson JM. 2010. Meat standards and grading: a world view. Meat Sci. 86:227–235, which underlines the importance of marbling. There traits might explain another axis.
On Figure 1, can you also indicate the projections of the main variables either in a separate graph or on the same graph?
Please, can you also provide a Figure for cluster analysis? It will show the distance between consumer groups.
For the discussion, I encourage you to comment and cite additional references:
Marie-Pierre Ellies-Oury, Alexandre Lee, Hervé Jacob & Jean-François Hocquette (2019) Meat consumption – what French consumers feel about the quality of beef?, Italian Journal of Animal Science, 18:1, 646-656, DOI: 10.1080/1828051X.2018.1551072 = this reference confirms more or less the importance of intrinsic traits saying that “inconsistency in eating quality” is important.
Henchion MM, McCarthy M, Resconi VC. 2017. Beef quality attributes: a systematic review of consumer perspectives. Meat Sci. 128:1–7. = this is a potential review paper for you.
Author Response
|
Reviewer’s question/comment |
Response/Rebuttal |
|
Lines 37-44. Please indicate the size of the market by providing, for instance, the number of people living in Venezuela, and the amount of beef produced within the country and the amount of imported beef. |
Agree, although it was, and still is, a challenging task to find reliable, updated information from governmental sources in Venezuela. Also, we felt the need for describing the main beef marketing channels in the country. Accordingly, we tried to please the reviewer's request in three manners. (1) By using a FAS-USDA-Gain report [3] in the Introduction that reports historical data (2000-2019), collected from private sources, indicating changes in market size (i.e., domestic production and importations, and per capita consumption) (2) Also, we have inserted paragraphs containing recent and projected data on the human population. (3) Lastly, in M&M (page 3, lines 99-102) we provided values (in MT) of the domestic beef production, imports, and human population for the two years of data collection (2007-2008) according to FAS-USDA-Gain Report (2019) and INE (2019). Changes in the Introduction (lines 37-57) follow: “Meat is regarded as the most valuable livestock product [1]. Its consumption remains relatively steady in the developed world; however, in developing countries, its annual per capita consumption has doubled since 1980 [1]. This is not the case in Venezuela where there is currently a strong contraction in the demand for beef due to rampant hyperinflation and a drastic loss of purchasing power [2,3,4]. The Venezuelan beef market has three main marketing channels: “Traditional” (local butcher shops that represents 60 percent of the market) which offers beef and beef products of different qualities, depending on location and the surrounding community’s economic circumstance; “Modern” (supermarkets and medium-sized grocery stores, selling packaged, higher quality that represents 30 percent of the market); and “Industrial” representing 10 percent of the market and is comprised of beef renderers and packers [3,4]. Higher oil prices during the first decade of the millennium allowed for subsidies to import beef products and(or) slaughter cattle from the USA, Brazil, Argentina, and Nicaragua; however, in 2003 Venezuela banned all US beef and beef products because of BSE regulatory concerns [3]. Domestic beef production, imports, and per capita consumption from 2000 to 2019 have been estimated from different private sources [4]. The beef per capita consumption that experienced a sustained rise during the period 2005 to 2011 and reached a historic peak of 24 kg in 2011, fell dramatically to 7 kg (a 70 % decrease) in 2018 [3]. Also, a lower domestic production (ca. 30TM) and no imports were estimated for 2019, given the deteriorated macroeconomic and market conditions [4]. The population of Venezuela could reach 28.7 million inhabitants by July 2020, a reduction of 7.42% (ca. 31 million) from the population estimated in 2018 [5]. This anticipated population contraction could be explained by the publicly known refugee crisis in the country [6].” Changes in M&M (lines 99-102) follow: “It is worth noting that the domestic beef production in 2007 and 2008 were 490 MT and 400 MT, respectively; whereas the beef imports for the same years were 290 MT and 380 MT, respectively [3]. It is also noteworthy that the Venezuelan population for 2007 and 2008 was 27.2 million and 27.7 million inhabitants, respectively [9].” |
|
The English language should be checked. |
The entire manuscript was carefully peer-revised in this regard. We feel we corrected all punctuation and grammar mistakes to improve the writing of the manuscript. |
|
|
|
|
For instance, line 87, it should be “selection of participants” |
All minor corrections were performed
|
|
Line 251, it should be intrinsic attributes (with “s”). |
|
|
|
|
|
Please could you indicate the difference between Conjoint Analysis (CA) and Factorial Analysis (FA) of Correspondence (which I would have recommended)? |
We acknowledge a typo here. The multivariate statistical analysis used in this study was Factor Analysis (FA). We run both exploratory and confirmatory FA. We apologize for this mistake and really appreciate the opportunity to correct it. Comment. Both CA and FA are reported to be used in Consumer´s perception studies. The CA is a statistical technique used in market research to determine how people value different features that make up an individual product or service; meanwhile, FA is generally used to examine how underlying constructs influence the responses on a number of measured variables. Both types of factor analyses are based on the Common Factor Model. |
|
|
|
|
Line 205: please define KMO. |
KMO stands for Kaiser-Meyer-Olkin. KMO is a parameter from the FA analysis that allows to exclude those variables with little weight on each Factor in a preliminary FA. The definition of KMO was included in the first paragraph in section 3.2 of Results. |
|
|
|
|
I think it is very surprising that marbling and butcher trustworthiness were removed as they are known as important traits for consumers. See for instance Polkinghorne RJ, Thompson JM. 2010. Meat standards and grading: a world view. Meat Sci. 86:227– 235, which underlines the importance of marbling. These traits might explain another axis. |
|
|
|
|
|
On Figure 1, can you also indicate the projections of the main variables either in a separate graph or on the same graph? |
We included the figure in a separate graph. We had to change the order of the Figures (see pages 11 and 12) |
|
|
|
|
Please, can you also provide a Figure for cluster analysis? It will show the distance between consumer groups. |
We accept Reviewer´s suggestion, however, we decided to use the Herarchical Clustering + Projection FA instead of the cluster dendrogram (see Figure 3, page 13 ) since it is more informative than the dendrogram. |
|
|
We thank for the references provided by the Reviewer, all of them were very helpful and included in the discussion section.
|
|
For the discussion, I encourage you to comment and cite additional references: Marie-Pierre Ellies-Oury, Alexandre Lee, Hervé Jacob & Jean-François Hocquette (2019) Meat consumption – what French consumers feel about the quality of beef?, Italian Journal of Animal Science, 18:1, 646- 656, DOI: 10.1080/1828051X.2018.1551072. This reference confirms more or less the importance of intrinsic traits saying that “inconsistency in eating quality” is important. Henchion MM, McCarthy M, Resconi VC. 2017. Beef quality attributes a systematic review of consumer perspectives. Meat Sci. 128:1–7. This is a potential review paper for you. |
Reviewer 2 Report
Comments to the Author:
The manuscript reports research into attitudinal determinants of beef consumption in Venezuela. The subject matter is interesting and falls into the scope of the journal.
Lines 23 and 73: “45 questions”, however, according to the description of the five sections (lines 73-78), the total questions are 35. Is there missing information or is there a mistake in the number “45”? Please, clarify this issue.
Line 81: could you explain why the number of consumers from Central region was higher (327) than those from Western or Eastern? Could the different number in each region have influenced in the final results? Explain this, please.
Line 109: “12].” instead of “12]”
Line 136: “Venezuela, would you prefer” instead of “Venezuela, you would you prefer”
The lines 136-140 would be better after line 109 because they refer to the intrinsic and extrinsic attributes.
Line 168: “does not” instead of “do not”.
Line 186: “for all variables (P < 0.05) but civil status (P = 0.14)” instead of “for all variables but civil status (P < 0.05)”.
Comments for all tables:
When one table is in two pages, the first row of the table should appear at the top of the second page. The term “P-value” used to express the statistical differences must be homogenized in the manuscript. This means that it should be in the same way in the manuscript. For instance now, it appears in the table 1 like “P-value” and in the table 2 like “P value”. When the P-value is lower than 0.001, write “P<0.001” instead of “P<0.000001”.
Table 1: check the column “Total, %”. There are several mistakes related to the sum. For instance, student = 6.3 instead of student = 6.2; full-time employee = 25.7 instead of full-time employee = 25.6; others = 4.4 instead of others = 4.5; more than 4 members = 34.2 instead of more than 4 members = 34.5; 4 members = 24.9 instead of 4 members = 24.8.
Table 2: check the column “Total, %”. There are several mistakes related to the sum. For instance, there are mistakes in “I like it” (38.4 instead of 37.9), “only if I have no other option” (2.8 instead of 2.7), “I do not like it” (0.3 instead of 0.4), “Rarely” (4.2 instead of 4.3), “once a week” (26.3 instead of 26.2), “decreased consumption” (22.1 instead of 22.2).
The term “P” used to express the statistical differences must be homogenized in the manuscript. This means that it should be in the same way in the manuscript. For instance now, in the lines 186, 200 and 218 it is expressed in different ways.
Line 223: “therefore, they were willing” instead of “therefore, were willing”.
Line 224: “Consumers from the” instead of “Consumers form the”.
Table 3: “Would you pay more for a tasty beef?” instead of “Would you pay more for a tasty beef”.
A paragraph (lines 216-226) is repeated (lines 229-239), therefore, eliminate lines 216-226.
Line 243: “FA1 and FA2” instead of “F1 and F2”.
Figure 1: axis x (“FA1” instead of “FC1”) and axis y (“FA2” instead of “FC2”).
Lines 249-250: The number of the tables is incorrect. Therefore: “Tables 4 (intrinsic attributes), 5 (extrinsic attributes) and 6 (consumer´s buying/consumption motivations and perceptions).” instead of “Tables 3 (intrinsic attributes), 4 (extrinsic attributes) and 5 (consumer´s buying/consumption motivations and perceptions.”
Line 253: “those from G1” instead of “those form G1”.
Line 256: “hence, they are willing” instead of “hence, are willing”.
Line 260: “considers” instead of “consider”.
Line 262: “was” instead of “were”.
Line 264: “consumers from G4 were willing” instead of “they were willing”.
Line 266: “Table 4” instead of “Table 3”.
Line 269: “Table 5” instead of “Table 4”. Likewise, revise and correct the questions in this table, since there are several mistakes related to the punctuation.
Line 272: “Table 6” instead of “Table 5”. Likewise, revise and correct the questions in this table, since there are several mistakes related to the punctuation.
Line 291: “[18],” instead of “[18].”.
Line 310: “however,” instead of “however.”
Lines 312 and 313: there are several mistakes related to the punctuation.
Line 317: “decision,” instead “decision.”.
Line 319: “countries,” instead “countries.”.
Lines 324 and 325: there are several mistakes related to the punctuation.
Lines 327: there are several mistakes related to the punctuation.
Line 330: “was” instead of “were”.
Line 335: there is a mistake with the reference Villalobos.
Line 344: “had” instead of “has”.
Section “References”: this section should be revised and corrected because there are several mistakes related to the format. MDPI’s style for citations and references lists are widely based on the style used by the American Chemical Society. At least, there are mistakes in the references 1, 2, 8, 14, 15, 16, 17, 19, 21, 23, 24, 25 and 26.
Author Response
|
Reviewer’s question/comment |
Response/Rebuttal |
|
The manuscript reports research into attitudinal determinants of beef consumption in Venezuela. The subject matter is interesting and falls into the scope of the journal. |
The authors deeply appreciate all detailed observations corrections and suggestions provided by Reviewer 2. We did our best to make the necessary corrections to improve the manuscript. |
|
Lines 23 and 73: “45 questions”, however, according to the description of the five sections (lines 73-78), the total questions are 35. Is there missing information or is there a mistake in the number “45”? Please, clarify this issue. |
We must clarify that the questionnaire consisted of 45 questions. Of these, 7 questions corresponded to the socio-demographic aspects of the respondents (Section I) and 4 questions corresponded to the preference for beef and consumption habits (Section II). We recognize that Question # 8 of the questionnaire (which refers to the preference for meat) should not be included in Section I. The rest (34 questions) were classified as: Intrinsic attributes (n = 10, Section III); extrinsic attributes (n = 14, Section IV); and motivations for the purchase and / or consumption (n = 10, SecctionV). The referee counts 35 because there was a transcription error in Sect. IV (4 was written instead of 14) Lines 86-92 (new version) was rewritten as follows: The survey’s method was qualitative and consisted in the application of a structured, questionnaire containing 45 questions divided into five sections that included: (1) Socio-demographic characteristics of the participants (Section I, 7 questions); (2) Habits in beef consumption (Section II, 4 questions); (3) Criteria for assessing the quality of raw and cooked meat (intrinsic quality attributes (Section III, 10 questions); (4) Criteria for the evaluation of extrinsic quality attributes (Section IV, 14 questions) and, (5) Motivations for the purchase and /or consumption of beef (Section V, 10 questions). |
|
Line 81: could you explain why the number of consumers from Central region was higher (327) than those from Western or Eastern? Could the different numbers in each region have influenced in the final results? Explain this, please. |
The Central region includes the Capital city (Caracas) and this region concentrates approximately 25% of the population of the country. The number of observations for each region is weighted for its population size and by the number of questions in the survey, therefore, the number of respondents by region is appropriate to the number of the population and the number of questions in the survey (35). Also, surveys were performed in three different cities within each region to get a better representation of the population, under the same methodology.
Because the cluster analysis is exploratory and not inferential, the differences in the number of observations is not a limitation. On the other hand, the factor analysis of correspondence is more sensitive to differences in the number of observations in the case of finding a high frequency of atypical responses that can cause distortion in the relationships between the variables. However, this was not the case of our data. Therefore, we are confident that by having unified the data of all the regions, following the same methodology and the exploratory characteristic of the study could not have affected the results of the study.
|
|
Line 109: “12].” instead of “12]” |
All minor corrections were performed
|
|
Line 136: “Venezuela, would you prefer” instead of “Venezuela, you would you prefer” |
|
|
The lines 136-140 would be better after line 109 because they refer to the intrinsic and extrinsic attributes. |
|
|
Line 168: “does not” instead of “do not”. |
|
|
Line 186: “for all variables (P < 0.05) but civil status (P = 0.14)” instead of “for all variables but civil status (P < 0.05)”. |
|
|
Comments for all tables: |
|
|
When one table is in two pages, the first row of the table should appear at the top of the second page. The term “P-value” used to express the statistical differences must be homogenized in the manuscript. This means that it should be in the same way in the manuscript. For instance, now, it appears in the table 1 like “P-value” and in the table 2 like “P value”. When the P-value is lower than 0.001, write “P<0.001” instead of “P<0.000001”. |
The term P value was carefully revised and homogenized in the entire manuscript. In Table 1 and Table 2, when the P-value is “P<0.000001, it was changed to “P<0.001” . |
|
|
|
|
Table 1: check the column “Total, %”. There are several mistakes related to the sum. For instance, student = 6.3 instead of student = 6.2; full-time employee = 25.7 instead of full- time employee = 25.6; others = 4.4 instead of others = 4.5; more than 4 members = 34.2 instead of more than 4 members = 34.5; 4 members = 24.9 instead of 4 members = 24.8. |
Tables 1 and 2 were corrected. The problem was that when total frequency values were adjusted to one decimal, the percentage sum in some cases exceeded 100%. After verified all the results from the contingency tables analysis to present the values with two decimals. |
|
Table 2: check the column “Total, %”. There are several mistakes related to the sum. For instance, there are mistakes in “I like it” (38.4 instead of 37.9), “only if I have no other option” (2.8 instead of 2.7), “I do not like it” (0.3 instead of 0.4), “Rarely” (4.2 instead of 4.3), “once a week” (26.3 instead of 26.2), “decreased consumption” (22.1 instead of 22.2). |
|
|
The term “P” used to express the statistical differences must be homogenized in the manuscript. This means that it should be in the same way in the manuscript. For instance, now, in the lines 186, 200 and 218 it is expressed in different ways. |
The term P value was revised in the entire manuscript and homogenized.
|
|
Line 223: “therefore, they were willing” instead of “therefore, were willing”. |
All minor corrections were performed
|
|
Line 224: “Consumers from the” instead of “Consumers form the”. |
|
|
Table 3: “Would you pay more for a tasty beef?” instead of “Would you pay more for a tasty beef”. |
|
|
A paragraph (lines 216-226) is repeated (lines 229-239), therefore, eliminate lines 216-226. |
|
|
Line 243: “FA1 and FA2” instead of “F1 and F2”. |
|
|
Figure 1: axis x (“FA1” instead of “FC1”) and axis y (“FA2” instead of “FC2”). |
|
|
Lines 249-250: The number of the tables is incorrect. Therefore: “Tables 4 (intrinsic attributes), 5 (extrinsic attributes) and 6 (consumer ́s buying/ consumption motivations and perceptions).” instead of “Tables 3 (intrinsic attributes), 4 (extrinsic attributes) and 5 (consumer ́s buying/consumption motivations and perceptions.” |
|
|
Line 253: “those from G1” instead of “those form G1”. |
|
|
Line 256: “hence, they are willing” instead of “hence, are willing”. |
|
|
Line 260: “considers” instead of “consider”. |
|
|
Line 262: “was” instead of “were”. |
|
|
Line 264: “consumers from G4 were willing” instead of “they were willing”. |
|
|
Line 266: “Table 4” instead of “Table 3”. |
|
|
Line 269: “Table 5” instead of “Table 4”. Likewise, revise and correct the questions in this table, since there are several mistakes related to the punctuation. |
|
|
Line 272: “Table 6” instead of “Table 5”. Likewise, revise and correct the questions in this table, since there are several mistakes related to the punctuation. |
|
|
Line 291: “[18],” instead of “[18].”Line 310: “however,” instead of “however.” |
|
|
Lines 312 and 313: there are several mistakes related to the punctuation. |
|
|
Line 317: “decision,” instead “decision.”. |
|
|
Line 319: “countries,” instead “countries.”. |
|
|
Lines 324 and 325: there are several mistakes related to the punctuation. |
|
|
Lines 327: there are several mistakes related to the punctuation. |
|
|
Line 330: “was” instead of “were”. |
|
|
Line 335: there is a mistake with the reference Villalobos. |
|
|
Line 344: “had” instead of “has”. |
|
|
Section “References”: this section should be revised and corrected because there are several mistakes related to the format. MDPI’s style for citations and references lists are widely based on the style used by the American Chemical Society. At least, there are mistakes in the references 1, 2, 8, 14, 15, 16, 17, 19, 21, 23, 24, 25 and 26. |
Each reference was revised and corrected according to MDPS´s style for citations and references list. Since we included new references to improve the Introduction and the Discussion sections, the entire Reference section was changed and adapted to the journal´s style. |